# Position: Biological Connectomes Must Be Interpreted as Forensic Artifacts of Learning, Not Blueprints for AI Architecture

## Abstract

This position paper argues that the recent push to "import the connectome" into machine learning architectures often rests on a flawed premise. We term this premise the Inverse Connectome Fallacy: the assumption that, because brains are intelligent, their wiring diagrams encode a general-purpose blueprint for intelligence in silicon. Connectomes are better understood as artifacts of optimization under biological constraints, including severe metabolic limits, three-dimensional wiring and delay costs, and biophysical noise. These constraints can strongly shape motifs such as sparsity, modularity, and small-world structure, which may not translate into advantages on modern accelerators. We propose reframing connectomes as forensic evidence: use them to test hypotheses about learning rules, objectives, and constraints that could have generated the observed structure. Concretely, we advocate convergence-style benchmarks, a graded substrate-alignment rubric for "connectome-inspired" claims, and stricter evaluation standards that privilege substrate-matched advantages over resemblance. We note that neuromorphic platforms like Loihi and SpiNNaker provide a principled setting to revisit structural motifs when cost models are intentionally closer to wetware.

---

**Position & Win Conditions**

**Claim.** "Connectome-inspired" architectural choices should be treated as *falsifiable* claims that require substrate-matched evidence, not as a default justification-by-resemblance.

**Definition.** We define a substrate-aligned advantage as a claimed benefit whose evaluation uses the same hardware, cost model, and constraints under which that benefit is supposed to matter (e.g., energy, latency, robustness, wiring, or noise).

**Scope taxonomy.** We separate three common aims: (i) *biological prediction* (models constrained by anatomy to explain biology), (ii) *substrate-aligned neuromorphic engineering* (structure studied under brain-like cost models), and (iii) *motif transfer to mainstream ML* (copying motifs into GPU/TPU training loops). Our critique targets (iii), and we support (i) and (ii) (Davies et al., 2018; Blouw et al., 2018; Ostrau et al., 2022; Yik et al., 2025).

**Win conditions.** A structural transfer claim is persuasive when it (i) states an intended advantage class (energy, latency, robustness, data-efficiency), (ii) evaluates that axis under a matched cost model, (iii) beats strong non-biomimetic baselines tuned comparably, and (iv) survives clean ablations that isolate the motif.

---

## 1. Introduction

**Position: structural biomimicry of connectomes, used as a default strategy for general-purpose ML on conventional accelerators, is unlikely to yield substrate-matched advantages and can waste compute absent unusually strong evidence.**

[1]Anonymous Institution, Anonymous City, Anonymous Region, Anonymous Country. Correspondence to: Anonymous Author <anon.email@domain.com>.

Preliminary work. Under review by the International Conference on Machine Learning (ICML). Do not distribute.

### 1.1. Why connectomes now feel actionable

The last few years have made connectomes feel newly actionable. Entire-brain wiring diagrams are no longer thought experiments but real artifacts, with unprecedented resolution across species and scales (Scheffer et al., 2020; Dorkenwald et al., 2024; Schlegel et al., 2024; MICrONS Consortium, 2025). A natural reaction in machine learning has followed: if intelligence emerges from neural circuits, it is tempting to treat those circuits as design templates for artificial neural network architectures. A visible line of NeuroAI work increasingly treats connectomes as architectural templates, graphs to be copied into silicon (Pircher et al., 2021; Bardozzo et al., 2023; 2024; Roberts et al., 2019; Lappalainen

et al., 2024).

**Exhibits: recurring "connectome-inspired" claim patterns.** A recurring pattern is that connectome- or biology-motivated wiring choices are justified by resemblance to biological circuitry, while the main quantitative evidence remains benchmark-centric (e.g., accuracy or task score), rather than the claimed advantage axis (energy/latency/robustness/substrate-matched efficiency). Representative examples include:

- **Connectome-as-architecture on standard benchmarks.** Full biological wiring diagrams are instantiated as fixed network topologies and evaluated primarily via classification accuracy (e.g., using *C. elegans* connectomes as image classifiers on MNIST-style tasks) (Park et al., 2023; Su et al., 2023).

- **Topology transfer framed as biological inspiration, evaluated mainly by accuracy/parameter efficiency.** Connectome-derived or connectome-motivated motifs are proposed as alternatives to standard designs and assessed largely via predictive performance or parameter counts, without directly measuring hardware-aligned metrics like energy or latency on a stated deployment substrate (Roberts et al., 2019; Bardozzo et al., 2024).

- **Connectome-constrained models for biological prediction.** Anatomy-constrained models can be scientifically valuable when the goal is to predict biological neural activity in the native organism (Lappalainen et al., 2024; Beiran & Litwin-Kumar, 2025).

These works are scientifically valuable, but they illustrate why "looks like a connectome" should not be treated as self-evidencing: without explicitly tying the claim to a target substrate and directly measuring the claimed advantage axis under a matched cost model, resemblance can substitute for the harder work of falsifiable, substrate-aligned evaluation.

### 1.2. A compact mini-survey of evaluation gaps (12 papers)

To make the concern empirical (without claiming exhaustiveness), we performed a lightweight mini-survey of 12 highly-cited, reputable, and representative papers (2019–2025) that (a) use connectomes as architectures or strong architectural priors, or (b) explicitly argue for connectome-derived inductive bias in ML-like tasks. We coded whether each paper *explicitly reports* key evaluation elements (axis, substrate, direct cost measurement, motif ablations, stress tests). Results are summarized in Table 1 and Figure 1. The main pattern is that *direct cost-model measurement* (energy/latency/wall-clock under declared deployment) is

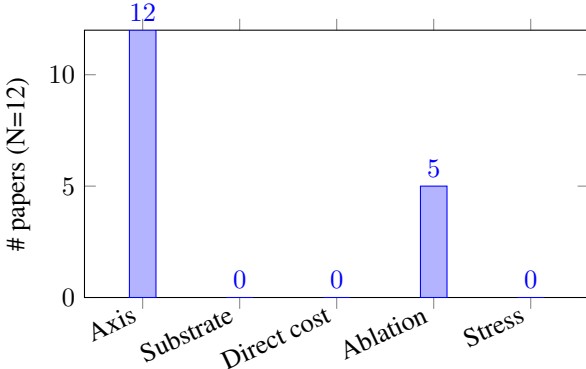

*Figure 1.* Mini-survey summary counts from Table 1. The primary gap motivating this position paper is not that connectome-inspired work is "wrong," but that it often lacks explicit substrate-aligned evaluation when making transferable-advantage claims.

rarely reported, while accuracy-centric evaluation and partial motif ablations are more common.

### 1.3. The Inverse Connectome Fallacy

The instinct to treat connectomes as default design templates is systematically risky as a route to general-purpose ML. We call the mistake the *Inverse Connectome Fallacy*: the assumption that a static wiring diagram provides a privileged blueprint for building intelligent machines on modern hardware. The brain's connectome is a historically contingent solution to an optimization problem under wetware constraints: three-dimensional spatial embedding, metabolic scarcity, slow and noisy biophysics, and developmental path dependence.

Those constraints differ sharply from the constraints that shape modern accelerators, where dense linear algebra, large shared memories, and high-bandwidth interconnects dominate the performance envelope (Hooker, 2020; Vaswani et al., 2017). Treating biological geometry as architectural wisdom risks importing biological *taxes* rather than biological *algorithms*.

The deeper point is not merely that "connectome-only" is incomplete. Neuroscience has repeatedly shown that very different parameterizations and even distinct circuits can yield similar dynamics and behaviors, a form of degeneracy that weakens any simple mapping from structure to function (Prinz et al., 2004; Edelman & Gally, 2001; Marder & Goaillard, 2006). If structure underdetermines function even in biology, then copying structure into a foreign substrate provides no guarantee of importing the relevant computation.

### 1.4. What the fallacy looks like in practice

The fallacy is rarely stated explicitly. Instead, it appears as recurring evaluation patterns. We list common failure modes and the corresponding corrective question, grounded

| Paper | Category | Stated axis | Axis | Substrate | Direct cost | Ablation | Stress test |
|---|---|---|---|---|---|---|---|
| Roberts et al. 2019 (Roberts et al., 2019) | Transfer | Accuracy | ✓ | – | – | – | – |
| Pircher et al. 2021 (Pircher et al., 2021) | Transfer | Accuracy/learning | ✓ | – | – | ✓ | – |
| Lanza et al. 2021 (Lanza et al., 2021) | Bio pred. | Behavioral fit | ✓ | – | – | – | – |
| Schmidgall et al. 2022 (Schmidgall et al., 2022) | Transfer | Architecture prior | ✓ | – | – | – | – |
| Barabási et al. 2023 (Barabási et al., 2023) | Transfer | Developmental prior | ✓ | – | – | – | – |
| Su et al. 2023 (Su et al., 2023) | Transfer | Params/"complexity" | ✓ | – | ✗[†] | – | – |
| Park et al. 2023 (Park et al., 2023) | Transfer | Accuracy | ✓ | – | – | – | – |
| Bardozzo et al. 2023 (Bardozzo et al., 2023) | Transfer | Accuracy | ✓ | – | – | ✓ | – |
| Suárez et al. 2024 (Suárez et al., 2024) | Forensic/bench | Task perf. + topology | ✓ | – | – | ✓ | – |
| Bardozzo et al. 2024 (Bardozzo et al., 2024) | Transfer | Accuracy | ✓ | – | – | ✓[*] | – |
| Yu et al. 2025 (Yu et al., 2025) | Transfer | Biofidelic arch. | ✓ | – | – | – | – |
| Costi et al. 2025 (Costi et al., 2025) | Transfer | Forecast error | ✓ | – | – | ✓ | – |

**Legend:** "Substrate" = an explicit intended deployment substrate/cost model (e.g., GPU/TPU vs neuromorphic) for the claimed advantage axis. "Direct cost" = direct reporting of energy/latency/wall-clock (or validated cost-model simulation) aligned with the stated advantage. [†] Parameter count is a proxy; it is not a direct energy/latency measurement. [*] Several connectome-architecture papers include rewiring/random controls; we credit ablation only when explicitly stated in accessible sections/abstracts.

*Table 1.* Mini-survey (N=12) of connectome-to-architecture and closely related papers: which evaluation elements are explicitly reported. This is not exhaustive, and is not meant to represent the entire field. However, it is a compact evidence point that accuracy-centric evaluation is common, while explicit substrate/cost-model evaluation is rare, among popular, highly-cited papers.

in a *graded* substrate-alignment rubric (Tables 2–3).

- **Resemblance-as-justification.** Structural similarity to a connectome motif is presented as primary evidence of likely ML advantage. *Corrective question:* What is the intended advantage axis, and where is it directly measured under a matched cost model?

- **Proxy substitution.** FLOPs, parameter count, or sparsity rate are used as substitutes for the claimed advantage (energy, latency, robustness, data-efficiency). *Corrective question:* Are energy and latency measured with a credible protocol, or are proxies used without calibration?

- **Baseline laundering.** The connectome-inspired model is compared against baselines that are under-tuned or mismatch the implementation substrate. *Corrective question:* Are baselines matched in tuning budget, optimization effort, and hardware-aware implementation?

- **Motif confounding.** Improvements are attributed to a motif, but other changes (capacity, regularization, optimizer) coincide with motif insertion. *Corrective question:* Is there an ablation that removes or randomizes the motif while holding capacity and training constant?

- **Axis drift.** The paper motivates a motif by one axis (e.g., efficiency) but evaluates mainly in-distribution accuracy. *Corrective question:* Are there stress tests aligned with the claimed axis (OOD for robustness; low-data for data-efficiency)?

- **Substrate mismatch by default.** The architecture is justified by biological constraints (3D embedding, wire length, spiking locality) but trained/deployed on dense-kernel GPU/TPU stacks. *Corrective question:* If the advantage depends on wetware costs, why should it transfer without a matched cost model or a neuromorphic substrate?

### 1.5. Scope and contributions

We clarify scope by separating three "connectome-inspired" aims: (i) *Biological prediction* (Lappalainen et al., 2024; Beiran & Litwin-Kumar, 2025); (ii) *Substrate-aligned neuromorphic engineering* (Davies et al., 2018; Davies, 2019; Yik et al., 2025); (iii) *Motif transfer to mainstream ML* (copying motifs into GPU/TPU training loops). Our critique targets (iii), and supports (i) and (ii).

We make three contributions. First, we articulate the Inverse Connectome Fallacy as a recognizable methodological error. Second, we argue that many celebrated connectome motifs plausibly arise from spatial and energetic constraints that do not carry over to mainstream silicon (Attwell & Laughlin, 2001; Chen et al., 2006; Bullmore & Sporns, 2012; Betzel et al., 2016; Hooker, 2020). Third, we propose (a) a graded substrate-alignment rubric and (b) convergence-style benchmarks that treat connectomes as test sets for hypothesized learning rules, along with reviewer and funding norms that make "connectome-inspired" a falsifiable claim class.

**Definition 1.1** (Substrate alignment (graded))**.** A connectome-inspired claim is *substrate-aligned* to the degree that (i) it specifies a target deployment substrate and advantage axis, and (ii) it evaluates that axis under a cost model that credibly approximates what is expensive on that substrate (energy/latency/memory movement/locality), with strong baselines and motif-isolating ablations.

*Figure 2.* Two ways to use connectomes. **Blueprint transfer** copies motifs as architectures and evaluates them on standard stacks, often using proxy metrics or mismatched baselines. **Forensic inference** treats connectomes as outcomes of learning under constraints and tests hypotheses by convergence of task performance and multi-scale structural agreement, under a substrate-alignment rubric.

## 2. Alternative Views

### 2.1. Evolution as architecture search

A credible opposing view claims evolution has performed a large-scale architecture search, and that ignoring biological architectures is misguided. If the brain achieves robust competence under stringent energy constraints, its wiring may encode inductive biases modern systems lack (Bullmore & Sporns, 2009; Sporns, 2011). We take this seriously, but it tightens (rather than relaxes) the evidentiary burden: if the claim is that a motif yields robustness or data efficiency, evaluation should prioritize those axes (stress tests, low-data regimes), not only average accuracy.

A second credible opposing view is pragmatic: even if connectome motifs are partly "taxes," they may regularize learning, stabilize recurrent credit assignment, or improve robustness. This motivates *which* stress tests and ablations should be standard in reviews (Tables 2–3).

### 2.2. Neuromorphic efficiency and the substrate argument

A second opposing view comes from neuromorphic engineering: the brain's energy efficiency may be inseparable from intelligence. Under this view, copying brain-like structure becomes essential for low-power AI. We do not deny

| Reviewer gate (minimal) | Minimum requirement for "connectome-inspired" architectural claims |
|---|---|
| State the axis | Specify the intended advantage (energy, latency, robustness, data-efficiency), not just accuracy. |
| Match the cost model | Measure the claimed axis directly when feasible; do not rely only on FLOPs/params as substitutes. |
| Strong baselines | Compare to non-biomimetic baselines that exploit the target substrate, with comparable tuning budgets. |
| Isolate the motif | Provide ablations that remove or randomize the motif while holding capacity and optimization constant. |
| Report failure modes | Include at least one stress test aligned with the claim (OOD for robustness; low-data for data-efficiency). |

*Table 2.* A reviewer-facing minimal gate. Passing the gate is necessary but not sufficient; Table 3 provides a graded alignment rubric.

this; we argue only that structure transfer should be evaluated under the *appropriate* cost model. Standardized neuromorphic benchmarking has repeatedly emphasized that comparisons can be misleading when interfaces/encodings/cost models are mismatched (Davies, 2019; Yik et al., 2025).

### 2.3. When blueprint transfer is expected to work

We narrow the claim to common practice: researchers (i) aim for general-purpose competence, (ii) deploy on GPUs/TPUs, and (iii) justify motifs primarily by resemblance rather than substrate-matched evaluation. Under those conditions, copying biological geometry is a high-risk bet.

We explicitly endorse connectome use in two settings. First, connectome-constrained models can be invaluable for biological prediction (Lappalainen et al., 2024). Second, blueprint transfer is plausibly expected to work when (a) the substrate enforces brain-like cost models or (b) the objective matches biological constraints (strict energy/latency budgets, event-driven sensing, local learning) (Davies et al., 2018; Blouw et al., 2018). The precondition is *substrate alignment*, not resemblance.

## 3. Why copying connectome geometry onto silicon induces suboptimality

### 3.1. Hardware-aligned computation and the hardware lottery

Modern deep learning is a case study in architectures co-evolving with accelerators. Transformers rose largely because their core operations map cleanly onto parallel dense matrix multiplication and high-throughput memory systems (Vaswani et al., 2017). This is an example of the hardware lottery: ideas win disproportionately because they align with available compute substrates (Hooker, 2020). In this regime,

| Level | Operational rubric (what a reviewer should look for) |
|---|---|
| SA-0 Un-aligned | No deployment substrate/cost model is stated; evaluation is primarily accuracy/task score; any "efficiency" is proxy-only (params/FLOPs) with no calibration. |
| SA-1 Proxy-aligned | Axis is stated, but cost evidence remains proxy-only (params/FLOPs/sparsity) with no validated mapping to energy/latency; baseline implementations are not hardware-aware. |
| SA-2 Par-tially aligned | Substrate is stated and the paper includes at least one cost-model analysis (e.g., roofline/memory movement estimates, structured sparsity patterns, or validated simulator), plus reasonably matched baselines. |
| SA-3 Di-rectly aligned | Direct measurement of the claimed axis (energy/latency/wall-clock) on target hardware (or validated hardware simulator), plus motif ablations and tuning-budget-matched baselines. |
| SA-4 Strongly aligned | SA-3 plus (a) multiple workloads that exercise the claimed advantage, (b) robustness/sensitivity checks (encodings, sparsity structure, kernel choices), and (c) reproducibility artifacts enabling independent verification. |

*Table 3.* Graded substrate-alignment rubric (SA-0 to SA-4). This replaces a binary "gate" interpretation and clarifies acceptable evidence when direct neuromorphic hardware access is unavailable.

architectural choices that fight the substrate pay immediate penalties in throughput, scaling, and iteration speed.

### 3.2. Sparsity mismatch: biological sparsity vs accelerator-friendly sparsity

Structural biomimicry often asks silicon to pay wetware taxes. Locality and small-world structure reduce wire length in brains, but on GPUs can translate into irregular memory access and low arithmetic intensity. Modular heterogeneity fragments computation into small kernels, reducing accelerator efficiency.

The gap between biological sparsity and hardware-efficient sparsity is especially instructive. Brains exhibit extreme sparsity because spikes and long-range wiring are expensive. By contrast, unstructured sparsity is difficult to exploit efficiently on GPU-class hardware. Large-scale evaluations show that unstructured sparsity can yield compression without reliably translating into speedups absent specialized support (Gale et al., 2019). Modern sparse-training work emphasizes that real speedups typically require *structured* patterns aligned with hardware primitives, and explicitly notes that unstructured sparsity is challenging to accelerate in practice (Lasby et al., 2024). This is the core mismatch: biological motifs often correspond to irregular sparsity and heterogeneous microstructure, while silicon-based speed favors structured computation.

### 3.3. What would count as evidence (and what does not)
Comparisons of connectome-derived architectures against conventional networks suggest that structural similarity

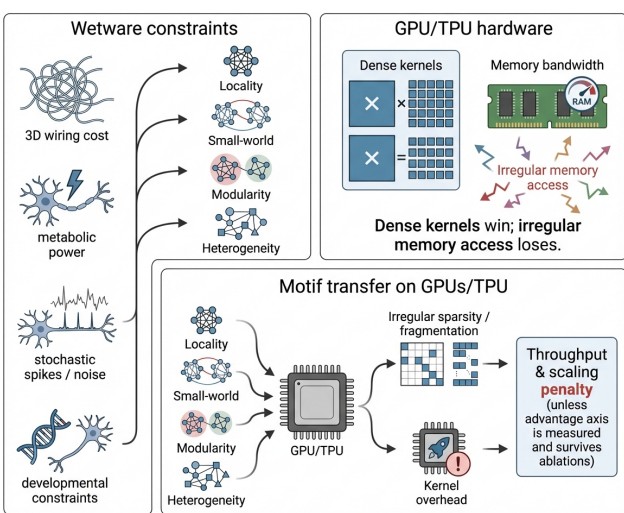

*Figure 3.* Motif transfer can fail under substrate mismatch...

alone does not guarantee improved performance (Pircher et al., 2021; Bardozzo et al., 2023; 2024). These results do not prove connectome-inspired models can never help, but they undermine the rhetorical move "it looks like the brain, therefore it should learn better." In practice, the claim becomes an ML contribution when it specifies an advantage axis and demonstrates that advantage under a credible, substrate-matched protocol against strong baselines and motif-isolating ablations (Tables 2–3).

## 4. Loihi and the substrate-alignment rubric

### 4.1. Why Loihi matters for this debate

Neuromorphic computing provides the strongest case for architectural biomimicry, and it is exactly why the right critique cannot be "never use spikes" or "never use connectomes." Intel's Loihi is a neuromorphic manycore processor designed for event-driven spiking computation and on-chip learning, targeting regimes where sparse activity and local updates can reduce energy (Davies et al., 2018). In application benchmarks such as keyword spotting, Loihi has shown substantially lower energy per inference than conventional hardware in certain low-power settings (Blouw et al., 2018). Broader benchmarking surveys emphasize that neuromorphic advantages depend on workload, software stack, and careful accounting of energy and latency (Ostrau et al., 2022).

### 4.2. Benchmarking discipline as a methodological constraint

The neuromorphic community has argued that claims of advantage must be benchmarked with discipline. Calls for standardized, application-relevant benchmarks emphasize that gains can be overstated when comparisons mix interfaces, encodings, or cost models (Davies, 2019). Recent commu-

nity efforts operationalize this point by providing bench-marking frameworks that systematically measure both algorithmic performance and hardware-relevant metrics (Yik et al., 2025). This clarifies what a strong biomimicry claim should look like: not "this graph resembles cortex," but "this structure yields measurable energy/latency benefits under a substrate whose cost model matches the hypothesized biological constraint."

### 4.3. Using the graded rubric in practice

The goal of SA-0 to SA-4 is to prevent the rubric from being interpreted as a veto: authors without neuromorphic access can still achieve SA-2 by providing credible cost-model analyses and substrate-aware baselines. Conversely, if a paper claims energy/latency advantages but reports only proxy metrics, it should not be evaluated as if it demonstrated SA-3/SA-4.

## 5. A forensic framework: from copying graphs to inferring objectives

### 5.1. Generative models as a template for inverse questions

If copying the connectome is the wrong question, what is the right one? The right question is inverse: what objectives, learning rules, and constraints generate connectome-like structure across datasets and scales? Network neuroscience already contains the beginnings of this program. Generative models treat a connectome as the output of wiring rules that trade off geometric cost against topological value, then fit those rules to match observed statistics (Betzel et al., 2016; Betzel & Bassett, 2017; Akarca et al., 2021; Rubinov et al., 2015).

A minimal instantiation is:

$$P(i \leftrightarrow j) \propto \exp(\alpha\, T_{ij} - \beta\, d_{ij})\,, \qquad (1)$$

where $d_{ij}$ is spatial distance and $T_{ij}$ is a topological "value" term.

### 5.2. From wiring rules to learning rules

Machine learning can broaden the meaning of "wiring rules." In biology, wiring emerges from developmental programs, local plasticity, homeostatic regulation, and reward-modulated learning. In ML terms, it is the result of an optimization process subject to constraints. A forensic NeuroAI agenda proposes candidate objectives and learning mechanisms, runs them in simulated developmental environments, and asks whether resulting networks converge to empirical connectome statistics.

This framing is more falsifiable than motif transfer. If a proposed learning rule cannot reproduce even coarse con-

nectome statistics under plausible constraints, it is missing something essential. Conversely, if a simple rule repeatedly produces the correct motifs across organisms and scales, that rule is a candidate transferable principle (even if the final wiring diagram is not). Recent work on wiring-aware objectives in artificial networks supports this bridge: incorporating wiring-economy terms can systematically change learned connectivity and performance (Zhang et al., 2025).

### 5.3. Connectomes become most valuable when paired with functional targets

The strongest evidence for the forensic view comes from settings where connectome information constrains models to explain biological responses, rather than serving as a generic architecture template. For example, connectome-constrained networks can predict neural activity across the fly visual system (Lappalainen et al., 2024). Functional connectomics combines anatomy with physiology and behavior, strengthening the link between structure and dynamics (MICrONS Consortium, 2025). That combination is what a forensic agenda needs: it prevents static graphs from being treated as complete explanations.

## 6. Convergence benchmarks and evaluation criteria

### 6.1. What a convergence benchmark measures

We propose that connectome-inspired ML adopt a benchmark culture that rewards explanatory convergence rather than superficial resemblance. A convergence benchmark starts with (i) a learning rule/objective class, (ii) constraints (energy, wiring cost, latency, locality), and (iii) a developmental or task environment. The benchmark score is not simply task accuracy; it is a joint score combining task competence and agreement between emergent network statistics and empirical connectomes. We formalize this idea by defining a convergence score that jointly evaluates functional competence and structural agreement. The score rewards task performance while penalizing deviations between statistics of the generated network and those observed in empirical connectomes, weighted by their explanatory importance:

$$S = S_{\text{task}} - \sum_k w_k\, d(\phi_k(G_{\text{gen}}), \phi_k(G_{\text{real}}))\,. \qquad (2)$$

### 6.2. A minimal "v1" instantiation

A minimal benchmark can fix a small, auditable set of statistics and distances: (i) edge-length/locality profile (Chen et al., 2006; Bullmore & Sporns, 2012); (ii) degree/weight distributions (Rubinov et al., 2015); (iii) motif counts (Bull-

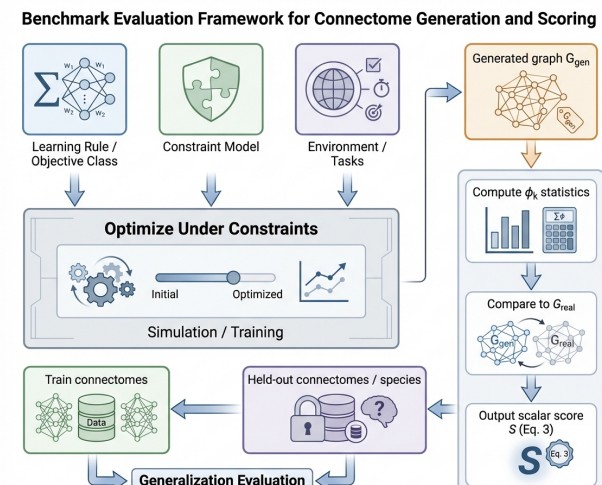

*Figure 4.* Benchmark evaluation framework for connectome generation and scoring. A learning rule/objective class, constraint model, and environment/task suite define an optimize-under-constraints simulation that produces a generated graph $G_{\text{gen}}$, which is scored by computing statistics $\Phi$ and comparing to a target connectome $G_{\text{real}}$ to yield a scalar $S(G_{\text{gen}})$, with optional generalization evaluation on held-out connectomes or species.

more & Sporns, 2009); (iv) modular organization (Sporns, 2011); (v) hubness/rich-club curves (Bullmore & Sporns, 2012). Distances $d(\cdot, \cdot)$ should be distributional distances (curves/histograms) to reduce single-number gaming. The benchmark must publish: statistics, distances, weights $w_k$, and uncertainty handling across reconstructions.

### 6.3. Targets across scales and datasets

The evaluation should discourage overfitting to a single connectome. A stronger target is partial invariance across species and labs. A practical protocol can be explicit: fit on a subset of connectomes (or regions), evaluate on held-out reconstructions/species, and report both task and structural generalization. Whole-brain fly datasets provide an immediate playground (Dorkenwald et al., 2024; Schlegel et al., 2024; Lin et al., 2024). Multimodal mouse datasets add functional grounding (MICrONS Consortium, 2025).

### 6.4. Failure modes the benchmark should detect

A convergence benchmark should score (and penalize): (i) *single-axis overfitting*; (ii) *dataset memorization*; (iii) *mechanism–cost mismatch*; (iv) *proxy substitution*. Treating these explicitly moves the field from narrative to hypothesis testing.

### 6.5. A concrete implementable benchmark proposal: *NeuroBench-Connectome Track (v1)*

To make this immediately actionable, we propose a "NeuroBench-Connectome Track" built on NeuroBench's

existing measurement discipline (Yik et al., 2025) and motivated by benchmarking cautions in neuromorphic computing (Davies, 2019). The track would:

- **Admit connectome-derived candidates** (e.g., connectome-based reservoirs (Suárez et al., 2024; Costi et al., 2025), connectome-to-architecture networks (Bardozzo et al., 2024; 2023; Park et al., 2023; Su et al., 2023)).

- **Require an explicit advantage claim** (energy, latency, robustness, data-efficiency) and a declared deployment substrate.

- **Report direct measurements where possible** (energy/inference, latency, throughput) using NeuroBench protocols; if hardware access is unavailable, require SA-2 cost-model analysis with validation checks.

- **Mandate motif-isolating ablations**: (i) rewired topology; (ii) topology-only vs weight-only hybrids (where applicable); (iii) capacity-matched baselines.

- **Include at least one stress test** aligned with the claim (OOD/shift; low-data; noise/perturbations for robustness).

This benchmark is deliberately "v1": it prioritizes standardized measurement and falsifiability over breadth. Over time it could add convergence scoring (Equation (2)) as a second axis for forensic claims.

## 7. Implications for reviewing and funding

### 7.1. Reviewing norms: make "connectome-inspired" falsifiable

The Inverse Connectome Fallacy persists partly because the community lacks shared evaluation norms. Reviewers should treat "connectome-inspired" as a claim that requires evidence, not a stylistic descriptor. If a paper argues an architecture merits attention because it mirrors connectome motifs, the review should ask what problem that motif solves on the target substrate, and whether the claim survives strong baselines and motif-isolating ablations under a matched cost model (Hooker, 2020). Absent that, the contribution is better framed as a hypothesis generator than as a demonstrated engineering advance.

### 7.2. Funding priorities and compute externalities

Connectomics and NeuroAI sit at a high-cost intersection. Whole-brain datasets, large-scale simulations, and specialized hardware can consume substantial resources. The opportunity cost of chasing the wrong abstraction is therefore unusually high. Funders should prioritize research that treats

connectomes as evidence for learning principles, especially when those principles can be tested across multiple datasets and aligned with realistic substrate constraints.

## 8. Call to Action

### 8.1. Raise the evidentiary bar for structural claims

Reviewers should treat "connectome-inspired" as a falsifiable claim. Papers that justify architectural choices primarily by resemblance should (i) state an intended advantage axis, and (ii) demonstrate it against strong baselines that exploit the target substrate, with (iii) motif-isolating ablations under comparable tuning. When the advantage is energy or latency, comparisons should report those quantities under credible protocols rather than relying on FLOPs or parameter count as substitutes.

### 8.2. Adopt a graded substrate-alignment rubric

Replace binary judgments with the SA-0–SA-4 rubric (Table 3). This discourages both extremes: it avoids granting "efficiency" claims undue credit when only proxies are reported, while also avoiding a de facto veto for authors without neuromorphic access.

### 8.3. Create convergence benchmarks that reward explanation

Benchmark builders should develop convergence-style evaluations where candidate learning rules/objectives/constraints are scored by reproducing connectome statistics as emergent outcomes, ideally while also achieving task competence in matched environments. Benchmarks should discourage overfitting to a single connectome by including multiple organisms or reconstructions, and treat mechanism–cost-model mismatch as a first-class failure mode.

## 9. Conclusion

### 9.1. Summary

High-resolution connectomes are a scientific triumph, but they create a temptation to confuse data with design. This paper argues that this temptation becomes a methodological error in ML when connectomes are treated as architectural blueprints on mismatched substrates. Connectome structure is shaped by wetware constraints and underdetermines function (Attwell & Laughlin, 2001; Chen et al., 2006; Prinz et al., 2004; Marder & Goaillard, 2006; Scheffer & Meinertzhagen, 2021; Beiran & Litwin-Kumar, 2025). Copying those structures into mainstream silicon without substrate alignment can import biological taxes and impede scaling (Hooker, 2020). Neuromorphic hardware shows brain-like choices can make sense when the substrate changes, but it

underscores that substrate alignment is the precondition, not an afterthought (Davies et al., 2018; Davies, 2019; Yik et al., 2025).

### 9.2. Prediction

As connectome datasets become larger and more multimodal, the most valuable NeuroAI outcomes will look less like imported wiring diagrams and more like validated learning principles. Blueprint transfer will become persuasive to the extent that it repeatedly achieves SA-3/SA-4: clear advantage axis, matched cost model, strong baselines, and clean ablations.

## Impact Statement

This position paper concerns research methodology in NeuroAI and connectomics. Its intended impact is to improve scientific and engineering practice by discouraging claims of architectural merit based mainly on structural resemblance to biological wiring diagrams, and by encouraging more falsifiable, mechanism-driven approaches that treat connectomes as evidence about learning rules, objectives, and constraints. If adopted, the proposed shift in evaluation norms could reduce wasted computational and research effort.

Risks include misuse to dismiss biologically grounded work wholesale, including work where substrate constraints matter (neuromorphic systems) or where connectomes constrain biological explanation. We therefore argue against connectome-to-architecture translation as a default route to general-purpose ML on conventional hardware, not against neuroscience-guided ML in general.

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
