# OpenReview forum: "Position: The Inverse Connectome Fallacy: Connectomes Are Forensic Artifacts of Learning, Not Blueprints for AI"
_ICML.cc/2026/Position_Paper_Track — Submitted to ICML 2026 Position Paper Track_

### Official Review · Reviewer_bdo4 · 2026-03-06

**Significance:** 4
**Argument Clarity:** 3
**Rating:** 5
**Confidence:** 4

**Questions:**

1. I found the "axis" column of table 1 to be a bit confusing, as the "stated [axes]" are not commensurate. Can you clarify what you mean here? Also, Figure 1 simply repeats information from Table 1, so I think the authors might consider dropping it.

2. Table 3 is hard to read because of the linebreaks and formatting. Can you re-organize this to make it more legible?

3. Why does the caption of Figure 3 end with an ellipsis? Is it incomplete?

4. In Section 4, the discussion of neuromorphic hardware focuses entirely on Intel's Loihi chip. Can you either provide some justification for why this is the only chip design you mention? I think it's worth providing a broader view of the neuromorphic hardware landscape here.

5. Reading through the author's arguments regarding substrate alignment, it occurs to me that one could take this line of reasoning to the extreme and construct a specialized neuromorphic chip that includes connectomic constraints at the hardware level. Could you comment on this possibility?

6. In Section 5, the authors argue for a set of inverse modeling questions that center around asking what normative considerations could have given rise to the observed structure of connectomes. Here it would be useful to put forward some principles for benchmarking in this area, as the authors do in other sub-sections. In particular, I have in mind the importance of null models, as reviewed in a network neuroscience context by [Váša and Mišić](https://www.nature.com/articles/s41583-022-00601-9). This is vital since it's not clear to me what level of identifiability one should reasonably expect here.

7.  In Section 5.1, the authors include a few references on generative models for observed connectivity. I think here it would be useful to mention the older work on wiring length minimization by Chklovskii and colleagues in the early 2000s. This focused to a great extent on *C. elegans*, for instance [Chen, Hall, and Chklovskii (2006)](https://www.pnas.org/doi/10.1073/pnas.0506806103).

8. In Section 5.3, the authors refer to the work of Lappalinen et al. as a successful prediction of activity across the fly visual system, but as far as I know those authors relied upon existing functional data (much of which was already published). It would be useful for the authors to comment on what they view as a sufficient standard for true predictions; should one declare success only if a model predicts new data?

9. In that vein, I think it would be useful to take a position on benchmarking standards for comparisons between models and neural data. For example, [Creamer et al.](https://www.biorxiv.org/content/10.1101/2024.09.22.614271v3) (uncited in the manuscript) measured predictive improvement for *C. elegans* data with connectomic constraints in terms of a normalized correlation that excludes substantial inter-animal variability in responses. For what level of prediction do the authors think one should aim?

10. It might be useful to mention work from Zador and colleagues on genomic bottlenecks, where information about the connectome serves as some form of intermediate compressed representation. Here I am thinking of work as in [Shuvaev et al](https://www.pnas.org/doi/10.1073/pnas.2409160121).

**Alternative Views Section:**

Yes

**Compliance With Llm Reviewing Policy A Conservative:**

Affirmed.

**Discussion Potential:**

4

**Final Justification:**

I was in favor of acceptance before the revision, and I believe the changes promised by the authors will further strengthen the paper.

**Paper Summary:**

This paper argues that the strategy of directly importing connectomes from biology to modern silicon hardware as a constraint/architectural bias for machine learning is fundamentally misguided. It outlines three aims of connectome-inspired architectures - biological prediction, substrate-aligned neuromorphic engineering, and motif transfer to mainstream ML - and mounts a detailed critique of the third.

**Position:**

Yes

**Position In Title:**

Yes

**Related Work:**

3

**Strengths And Weaknesses:**

The paper makes a generally well-articulated argument in favor of its stated position. The writing is mostly clear, though in some places I think the flow could be improved by smoothing transitions between the paper's many short sub-sections. On the whole I am in favor of acceptance, as I think the paper has good potential to spark useful discussion in the ICML community, particularly the subset of the community interested in NeuroAI.

To avoid repeating myself, I list most of my specific concerns and suggestions under **Questions**. I would like to highlight two specific issues here:

1. The authors clearly (and I think usefully) distinguish three distinct aims of connectome-inspired modeling in Section 1. However, the remainder of the paper is not always organized in a way that reflects these distinct aims. I would suggest that the authors structure the paper to directly reflect these distinct aims as clearly as possible.
2. One component of the paper's position is that connectome-inspired neuromorphic computing is a promising application of connectome data. However, the discussion of this point focuses almost entirely on Intel's Loihi chip (Section 4). This (a) does not provide a balanced view of the neuromorphic hardware landscape and (b) also excludes the possibility of connectome-guided hardware design, which which occurs to me as the logical extreme of the idea of substrate alignment.

If the authors can address these issues, and the others which I note below, I think this paper can make a useful contribution to the discussion around connectomics and NeuroAI.

**Support:**

3

---

> ### Author Rebuttal · Authors · 2026-03-28
>
> Thank you for the careful and constructive review. We appreciate the specific suggestions for strengthening the paper and address each below.
>
> **On organization around the three aims**: We agree that the manuscript should more explicitly track the three aims introduced in Section 1: biological prediction, substrate-aligned neuromorphic engineering, and motif transfer to mainstream ML. In revision, we will strengthen the manuscript's signposting and restructure portions of the discussion so that the critique of aim (iii) remains visibly separated from our support for aims (i) and (ii).
>
> **On the Loihi-only focus in Section 4**: We focused on Loihi because it has the most detailed publicly available benchmarking data for the substrate-alignment argument (Davies et al., 2018; Blouw et al., 2018; Davies, 2019), particularly regarding on-chip learning and energy accounting. However, the reviewer is right that the section should not suggest Loihi is the only relevant platform. The abstract mentions SpiNNaker (Furber et al., 2013), but Section 4 does not discuss it. We will broaden Section 4 to include SpiNNaker and other relevant platforms, and discuss how different neuromorphic architectures (event-driven manycore, analog crossbar, mixed-signal) each impose distinct cost models that shape which connectome motifs are substrate-aligned.
>
> **On connectome-guided hardware design**: This is an insightful extension of the substrate-alignment logic. If the argument is that motifs should match the substrate, then designing the substrate to match the motifs is the natural complement. We view this as compatible with our position: the key requirement remains that the claimed advantages be evaluated under the actual cost model of the resulting hardware, rather than assumed from biological resemblance. We will add a brief discussion of this possibility.
>
> **On presentation details**: We agree that Figure 1 is redundant with Table 1 and will remove or merge it. The "stated axis" column in Table 1 reports each paper's stated intended advantage, which varies across papers; we will clarify the labeling. We will also reformat Table 3 for legibility and fix the ellipsis in the Figure 3 caption, which is a formatting artifact.
>
> **On null models in Section 5 (Vasa and Misic)**: We agree that null models are essential for interpreting convergence results; without them, it is unclear whether a learning rule's match to connectome statistics is meaningful or merely trivial. We will cite Vasa and Misic and discuss null-model benchmarking as part of the convergence framework.
>
> **On Chklovskii and wiring-length minimization in Section 5.1**: Chen, Hall, and Chklovskii (2006) is already cited in Sections 1.5 and 6.2, but we agree it deserves more prominent discussion in Section 5.1 as a foundational example of the inverse/forensic approach. We will move the citation and add a discussion there.
>
> **On prediction standards for Lappalainen et al. (Section 5.3)**: The reviewer is right that the key distinction is whether neural data were used during model fitting versus reserved for evaluation. We do not think success requires predicting entirely new data; prediction of held-out neural responses not used during fitting is sufficient, even if those responses were previously published. The critical requirement is a clean train/test separation, not novelty of the dataset. We will clarify this standard and make the prediction-versus-postdiction distinction an explicit evaluation criterion in the forensic framework.
>
> **On benchmarking standards for neural data (Creamer et al.)**: We will cite Creamer et al. and discuss their normalized-correlation approach as an example of a principled prediction metric that accounts for inter-animal variability. This connects directly to Section 6: convergence benchmarks should specify what level of neural prediction counts as success, not just structural match. We think the minimum standard should be prediction above a noise-corrected null (e.g., normalized correlation that factors out inter-animal variability, as in Creamer et al.), evaluated on held-out animals or conditions. The exact threshold will be dataset-dependent, but the key requirement is that it be stated explicitly and benchmarked against a connectome-free baseline, so the marginal value of connectomic constraints can be isolated.
>
> **On genomic bottlenecks (Zador, Shuvaev et al.)**: This is a valuable connection. The genomic bottleneck framework treats the transferable object less as a final wiring diagram and more as a compressed generative specification of circuitry, which aligns with our forensic framing: the important object is the process that gives rise to structure, not the final graph alone. We will add this reference and discussion.
>
> We welcome any follow-up questions during the discussion period.

---

> > ### Author Rebuttal · Reviewer_bdo4 · 2026-03-31
> >
> > Thank you for your detailed responses to my comments, which address most of the questions and concerns I had. I will maintain my recommendation of acceptance.

---

### Official Review · Reviewer_1mDa · 2026-03-08

**Significance:** 3
**Argument Clarity:** 3
**Rating:** 5
**Confidence:** 3

**Questions:**

- How robust would the conclusions of the mini-survey remain if applied to a much broader sample of connectome-inspired ML papers?

- Could the proposed substrate-alignment rubric be adapted for other forms of biological inspiration, such as synaptic plasticity rules or neuromodulation?

- How would the proposed convergence benchmarks handle scenarios where multiple valid generative models produce similar connectome statistics?

- Could this forensic approach to connectomes indirectly guide architecture design by helping to discover transferable principles?

**Alternative Views Section:**

Yes

**Compliance With Llm Reviewing Policy A Conservative:**

Affirmed.

**Discussion Potential:**

3

**Final Justification:**

The authors have addressed my concern after the rebuttal. I suggest to accept this paper

**Paper Summary:**

The paper introduces the concept of the Inverse Connectome Fallacy, arguing against the common practice of using biological brain connectomes as direct architectural blueprints for artificial neural networks. Instead, the authors propose that connectomes should be viewed as "forensic evidence" or historical artifacts of learning optimized under specific biological constraints (e.g., metabolic limits, wiring costs).

To support this position, the paper presents a mini-survey (N=12) highlighting evaluation gaps in current connectome-inspired ML research. It proposes actionable solutions, including a substrate-alignment evaluation rubric (SA-0 to SA-4) and convergence benchmarks to test if learning rules can reproduce connectome statistics.

**Position:**

Yes

**Position In Title:**

Yes

**Related Work:**

3

**Strengths And Weaknesses:**

Strengths
- Compelling Thesis: The "Inverse Connectome Fallacy" is a clear, memorable, and well-argued concept.

- High Significance & Timeliness: Addresses an important and expensive methodological issue in the growing field of NeuroAI.

- Constructive Approach: Instead of simply criticizing, the paper offers practical, actionable solutions (evaluation rubrics, benchmarking culture, reviewing norms).

- Balanced Argumentation: Effectively engages with alternative views (e.g., neuromorphic efficiency, evolution as architecture search) without resorting to strawman arguments.

- Good Contextual Awareness: Demonstrates a strong understanding of relevant interdisciplinary literature spanning connectomics, neuromorphic computing, and ML hardware constraints.

Weaknesses
- Limited Empirical Grounding: The mini-survey of 12 papers is relatively small and informal, which weakens the empirical support for the argument.

- Conceptual Benchmark Proposal: The suggested convergence benchmarks currently lack clear implementation details.

- Potential Overgeneralization: The authors may be overgeneralizing, as some existing connectome-inspired work already focuses on learning rules rather than direct architectural copying.

- Scope Drift: The paper occasionally shifts its focus between criticizing the copying of architectures and criticizing evaluation practices, which slightly impacts clarity.

**Support:**

3

---

> ### Author Rebuttal · Authors · 2026-03-28
>
> Thank you for the thoughtful review. We are glad the Inverse Connectome Fallacy, the constructive framing, and the balanced treatment of alternative views came through clearly. We address each weakness and question below.
>
> **On the mini-survey scope (N=12) (Q1)**: Our claim is illustrative rather than prevalence-estimating: the survey is used to demonstrate that the evaluation gap exists among prominent papers, not to estimate how common it is across the entire literature. The paper states this explicitly. Even within this sample, the finding that 0 of 12 report direct substrate-matched cost-model measurement is sufficient to motivate the need for clearer evaluation norms.
>
> **On benchmark implementation details**: We agree that a worked numerical example would make the proposal more concrete. Our intent was to specify a deliberately minimal v1 benchmark rather than only an abstract desideratum: Section 6.2 defines an initial statistic-and-distance set, and Section 6.5 proposes a concrete evaluation track with explicit claim statements, required ablations, and stress tests. We will clarify this framing and add a numerical walkthrough to the camera-ready version.
>
> **On overgeneralization**: We take care to separate three aims and direct the critique only at (iii) motif transfer to mainstream ML. Work that already focuses on learning rules rather than architectural copying falls under the forensic approach we advocate, and the paper explicitly credits such work (e.g., Betzel et al., 2016; Zhang et al., 2025 on wiring-economy objectives; Lappalainen et al., 2024 on connectome-constrained biological prediction). Our claim is not that all connectome-inspired work commits the fallacy, but that the field lacks shared norms to distinguish motif-transfer claims that do from those that do not. We will revise to make this distinction sharper.
>
> **On scope drift**: The paper is not making two unrelated critiques; it argues that weak evaluation norms allow blueprint-copying claims to survive. Section 1.4 maps each failure mode (resemblance-as-justification, proxy substitution, baseline laundering, etc.) to a specific evaluation gap. We will tighten the transitions to make this connection more explicit throughout.
>
> **On SA rubric generality (Q2)**: We believe the rubric logic extends naturally beyond connectome-inspired claims. The core structure (state the axis, match the cost model, strong baselines, isolate the motif, stress-test the claim) applies to any bio-inspired architectural claim, including those motivated by synaptic plasticity rules or neuromodulation. We will explicitly note this broader applicability.
>
> **On degeneracy/multiple valid generative models (Q3)**: If multiple learning rules produce statistically indistinguishable connectome outputs, a high convergence score identifies an equivalence class of objectives and constraints rather than a unique mechanism. This is informative: it shows that the structural outcome is robust to the choice of mechanism, which is itself a transferable insight. The benchmark should report the diversity of high-scoring rules, not just the top scorer, so that degeneracy is surfaced rather than hidden.
>
> **On forensic approach guiding architecture (Q4)**: Yes - this is exactly our intent. Section 5.2 argues that if a simple learning rule repeatedly produces the correct motifs across organisms and scales, that rule is a candidate transferable principle, even if the final wiring diagram is not. The forensic approach guides architecture indirectly: not by copying structure, but by identifying which objectives and constraints reliably generate useful structure, and then applying those objectives to the target substrate.
>
> We hope these clarifications address the main uncertainties and would welcome any follow-up questions during the discussion period.

---

> > ### Author Rebuttal · Reviewer_1mDa · 2026-04-01
> >
> > The authors' responses address most of the concerns I had. I have adjusted my recommendation accordingly. Please make sure the final version of the manuscript is revised by following the responses.

---

### Official Review · Reviewer_Vdfv · 2026-03-08

**Significance:** 3
**Argument Clarity:** 4
**Rating:** 5
**Confidence:** 3

**Questions:**

- How do the authors propose handling the degeneracy problem for the inverse question? If two very different learning rules produce statistically indistinguishable connectome outputs, what does a high convergence score actually tell us about transferable principles?
- SA-3 on Loihi: what would a credible energy/latency measurement protocol look like in practice, given the known difficulty of fair energy accounting on neuromorphic platforms (encoding costs, spike rate dependence, etc.)?

**Alternative Views Section:**

Yes

**Compliance With Llm Reviewing Policy A Conservative:**

Affirmed.

**Discussion Potential:**

4

**Final Justification:**

I maintain my initial rating of 5 Accept

**Paper Summary:**

Existing works consider the biological wiring of the brain as a blueprint for designing AI/ML models. In this paper, the authors argue against this, and say that connectomes are not a general blueprint for intelligence, rather, it is one solution because of biological constraints. Modern GPUs do not face the same constraints (it does matrix operations and fast memory), and hence AI/ML models should not follow connectome design. Furthermore, the authors present that existing AI/ML work present only accuracy metrics ; they do not present other metrics like cost/complexity, motif ablations, stress tests etc.

**Position:**

Yes

**Position In Title:**

Yes

**Related Work:**

3

**Strengths And Weaknesses:**

**Strengths**
- In the work, the authors propose to move away from biological connectomes to dictate AI/ML designs (blueprint transfer), and instead move towards forensic inference: to focus beyond task accuracy but also other axis like costs, motif ablations etc. Flipping the question from "what structure should I copy?" to "what learning rule, objective, and constraint set generates this structure?" is more scientifically tractable and likely to yield transferable principles.
- The authors surveyed existing works and found that all of them are in the realm of blueprint transfer. This motivates the authors' push for moving towards forensic inference instead.
- The Inverse Connectome Fallacy is well-named and useful, and will inspire discussion in the community.

**Weakness**
- Equation 2 : The score can be gamed by tuning $w_k$ post hoc, so it is unclear how to aggregate across species with different connectome sizes or reconstruction uncertainties. It would be great if there was atleast one numerical example.
- The authors say that unstructured biological sparsity if hard to accelerate on GPUs. Did the authors look into existing work that are structured sparse architectures with hardware-aligned speedups?

**Support:**

3

---

> ### Author Rebuttal · Authors · 2026-03-28
>
> Thank you for the detailed feedback. We are glad the forensic-inference reframing, the mini-survey motivation, and the Inverse Connectome Fallacy resonated. We address each point below.
>
> **On Equation 2, gaming and cross-species aggregation**: This is a fair concern. The benchmark deliberately uses distributional distances over curves and histograms (Section 6.2) rather than scalar summaries, thereby raising the cost of post hoc tuning. Cross-species holdout evaluation (Section 6.3) further limits overfitting: we propose fitting on a subset of connectomes and evaluating on held-out reconstructions or species, so a model tuned post hoc to one organism should score poorly on held-out targets. For aggregation across species with different sizes and reconstruction uncertainties, we envision per-species normalization of each distance (e.g., z-scoring against a random-wiring null model), making scores comparable without requiring identical connectome scales. We will add a concrete numerical walkthrough in the camera-ready version.
>
> **On structured sparsity**: Yes, we did look into structured sparse architectures with hardware-aligned speedups. The paper cites Lasby et al. (2024) on structured sparse training and Gale et al. (2019) on unstructured sparsity, and the argument is precisely that hardware-aligned speedups require structured patterns (block, N:M, channel-level), whereas biological connectome motifs typically correspond to irregular, unstructured sparsity. We are not arguing against sparsity in general, only that importing biological sparsity patterns onto GPUs without restructuring them to match hardware primitives is where the mismatch arises. We will sharpen this distinction in the text.
>
> **On the degeneracy problem**: If two very different learning rules produce statistically indistinguishable connectome statistics, a high convergence score tells us something about the equivalence class of objectives and constraints, not a unique mechanism. We view this as informative rather than problematic: it narrows the set of hypotheses, even without uniquely determining a single rule. If a broad family of rules all converge, that robustness is itself a transferable insight (the structure is not fragile to mechanism choice). If only a narrow class succeeds, that is stronger evidence for a specific principle. The benchmark should therefore report the diversity of high-scoring rules and not just the top scorer. We will clarify this interpretation in revision.
>
> **On SA-3 energy measurement on Loihi**: We agree this is practically difficult, which is exactly why the graded rubric exists. A credible SA-3 protocol should account for encoding/decoding costs (not just inference-core energy), spike-rate dependence (since energy scales with activity), idle/leakage power, and standardized input/output interfaces so comparisons are not confounded by different data pipelines. NeuroBench (Yik et al., 2025) and Davies (2019) provide starting points for this discipline. For authors without access to neuromorphic hardware, SA-2 remains a credible path through validated cost-model simulations or roofline analyses, with stated assumptions. The rubric is designed to ensure that the difficulty of SA-3 does not disqualify a claim, while simultaneously preventing proxy-only claims from being treated as direct measurements.

---

> > ### Author Rebuttal · Reviewer_Vdfv · 2026-04-03
> >
> > Authors address my questions

---

### Official Review · Reviewer_zP5h · 2026-03-13

**Significance:** 3
**Argument Clarity:** 4
**Rating:** 5
**Confidence:** 4

**Questions:**

- Mini-survey was indeed mini; are there lessons to be learned or claims about model architectures that could improved via evaluations proposed here outside of NeuroAI subareas?
    - Section 3 argues copying connectome onto silicon is suboptimal, but aren't there cases where there is empirical improvement (even by chance)? What can be done to forensically investigate those cases without requiring substrate matching?

**Alternative Views Section:**

Yes

**Compliance With Llm Reviewing Policy A Conservative:**

Affirmed.

**Discussion Potential:**

3

**Final Justification:**

Rebuttal has clarified some things but my overall opinion for acceptance remains unchanged.

**Paper Summary:**

This paper clearly lays out its position that treating connectomes as a strategy for AI in the absence of metrics evaluating their effectiveness (along advantage axes) is insufficient, and that such motif transfer must be more carefully evaluated. The paper is well written, clear, and convincing.

**Position:**

Yes

**Position In Title:**

Yes

**Related Work:**

3

**Strengths And Weaknesses:**

- Not sure the claim that using connectomes are a default strategy across ML is substantiated (for some NeuroAI, yes, but not broadly). But the overall position is well-argued and supported.
    - Figure 1 is largely superfluous (redundant with Table 1).
    - Other figures (diagram/schematics) are very well done, clear, and precise.
    - The inclusion of concrete criteria (for e.g. reviewing papers) is excellent.
    - The convergence benchmark idea as a joint score sounds novel and reasonable.

**Support:**

4

---

> ### Author Rebuttal · Authors · 2026-03-28
>
> Thank you for the thoughtful and encouraging review. We are glad the paper's central position, concrete reviewing criteria, and convergence-benchmark framing came across clearly. We address both questions below.
>
> **On the concern about over-breadth**: We agree that the strongest version of our claim targets a specific section of NeuroAI and not ML broadly. The intended scope is connectome-inspired motif transfer to mainstream GPU/TPU ML, where biological resemblance sometimes substitutes for matched evaluation of the claimed advantage axis. This is why the paper separates three aims and directs its critique only at (iii), while explicitly supporting (i) biological prediction and (ii) substrate-aligned neuromorphic engineering. We will revise the framing to make this scope unmistakable and will also remove/merge Fig 1 with Table 1 to remove redundancy.
>
> **On empirical improvements from copied motifs**: We agree these cases are important. Our position is not that such gains are impossible, but that they should be treated as hypotheses that require causal disambiguation. The graded SA rubric is designed exactly for this case. A copied motif that survives matched non-biomimetic baselines, rewiring/randomization controls, and a claim-aligned stress test would count as suggestive SA-1/SA-2 evidence, even if it didn't have a full substrate-matched energy/latency measurement. We will make this interpretation more explicit in revision.
>
> **On lessons beyond NeuroAI**: The same evaluation logic applies whenever an architectural motif is justified by a claimed benefit but evaluated mainly on headline accuracy, including sparsity/compression, sparse MoE efficiency claims, and robustness-oriented architecture proposals. We will add a sentence making this broader applicability explicit.

---

> > ### Author Rebuttal · Reviewer_zP5h · 2026-04-01
> >
> > My primary questions remain not fully answered, with authors largely promising to expand more in a revised manuscript. There is space in the rebuttal to provide concrete answers to my questions.

---

### Decision · Program_Chairs · 2026-04-30

**Decision:**

Reject

**Comment:**

The submission proposes the "Inverse Connectome Fallacy," arguing that using biological connectomes as direct blueprints for AI architectures is misguided. Instead, the authors advocate for forensic inference to uncover the biological optimization rules and constraints that generate observed neural structures. While reviewers acknowledged the timely conceptual contribution and clear taxonomy, the manuscript suffers from significant limitations.

The empirical grounding is notably thin, relying on a small, non-representative survey (N=12). Furthermore, the proposed benchmarking and evaluation framework lacks the necessary implementation details to be actionable, rendering the contribution more speculative than operational. Reviewers also identified problematic scope drift, where the argument inconsistently oscillates between architectural design and general evaluation norms. The opinions seem biased and fail to acknowledge the random exploration of architectures in AI and the major contributions of connectomics constraints. Given the lack of rigorous empirical validation, concrete methodological depth, and biased exposition, there does not seem much of a take home message.